Physiological responses of Oxyrrhis marina to a diet of virally infected Emiliania huxleyi

Goode Andrew G. 1 2
Fields David M. 1
Archer Stephen D. 1
Martínez Martínez Joaquín jmartinez@bigelow.org 1
1 Bigelow Laboratory for Ocean Sciences , East Boothbay , ME , United States of America
2 School of Marine Sciences, University of Maine , Orono , ME , United States of America
Breitbart Mya
Electronic publication date: 2019 Apr 19
Publication date: 2019
Volume: 7
Electronic Location ID: e6722
Received 2018 Apr 9; Accepted 2019 Mar 5
Copyright: ©2019 Goode et al.
Copyright year: 2019
Copyright holder: Goode et al.
License: This is an open access article distributed under the terms of the Creative Commons Attribution License, which permits unrestricted use, distribution, reproduction and adaptation in any medium and for any purpose provided that it is properly attributed. For attribution, the original author(s), title, publication source (PeerJ) and either DOI or URL of the article must be cited.
License URL: https://creativecommons.org/licenses/by/4.0/

Keywords: Phytoplankton, Emiliania huxleyi, Carbon, Virus, Grazing, Dinoflagellate, Food web, Zooplankton, Growth, Oxyrrhis marina, Coccolithophore

Funding: National Science Foundation Grant EAR 1460861 REU Site: Bigelow Laboratory for Ocean Sciences–Undergraduate Research Experience in the Gulf of Maine and the World Ocean Gordon and Betty Moore Foundation Grant GBMF3397 Bigelow Laboratory for Ocean Sciences and the University of Maine internal funding This research was supported by funds from the National Science Foundation through Grant EAR 1460861–REU Site: Bigelow Laboratory for Ocean Sciences–Undergraduate Research Experience in the Gulf of Maine and the World Ocean, Gordon and Betty Moore Foundation Grant GBMF3397, and by the Bigelow Laboratory for Ocean Sciences and the University of Maine internal funding. The funders had no role in study design, data collection and analysis, decision to publish, or preparation of the manuscript.

==============================
The coccolithophore Emiliania huxleyi forms some of the largest phytoplankton blooms in the ocean. The rapid demise of these blooms has been linked to viral infections. E. huxleyi abundance, distribution, and nutritional status make them an important food source for the heterotrophic protists which are classified as microzooplankton in marine food webs. In this study we investigated the fate of E. huxleyi (CCMP 374) infected with virus strain EhV-86 in a simple predator-prey interaction. The ingestion rates of Oxyrrhis marina were significantly lower (between 26.9 and 50.4%) when fed virus-infected E. huxleyi cells compared to non-infected cells. Despite the lower ingestion rates, O. marina showed significantly higher growth rates (between 30 and 91.3%) when fed infected E. huxleyi cells, suggesting higher nutritional value and/or greater assimilation of infected E. huxleyi cells. No significant differences were found in O. marina cell volumes or fatty acids profiles. These results show that virally infected E. huxleyi support higher growth rates of single celled heterotrophs and in addition to the “viral shunt” hypothesis, viral infections may also divert more carbon to mesozooplankton grazers.

Introduction

Cell lysis, due to viral infection, accounts for up to 30% of daily mortality rates of marine microorganisms (Suttle, 1994; Van Hannen et al., 1999), influences shifts in microbial community structure (Martínez Martínez et al., 2006; Thingstad, 2000), and is hypothesized to result in the reduction of infected eukaryotic phytoplankton’s net primary productivity (Suttle, 1994) while increasing the overall ecosystem’s net primary productivity (Weitz et al., 2015). Conventional dogma holds that virus-induced cell lysis diverts energy away from the traditional food web by releasing the organic carbon (C), nitrogen (N), and phosphorus (P) in phytoplankton cells to the dissolved phase, fueling an active bacterial population. This process, known as the “viral shunt”, is hypothesized to transfer 6–26% of C (estimated 150 gigatons of C per year) from photosynthetic plankton to the dissolved organic pool (Suttle, 2005; Wilhelm & Suttle, 1999). However, to the best of our knowledge, the magnitude of the C “shunt” during viral infection has not been directly measured. A quantitative understanding of the pathways and factors that affect the flow of organic C in marine systems is key to understanding community structure and for predicting resource availability to support important commercial species. Although it is known that viral infection of algal cells alters crucial cellular and biogeochemical processes (Evans, Pond & Wilson, 2009; Gilg et al., 2016; Malitsky et al., 2016; Rosenwasser et al., 2014; Suzuki & Suzuki, 2006), the impacts of these changes on the nutritional value of cells and on the grazing and growth rates of both micro- and macrozooplankton are largely unexplored (Evans & Wilson, 2008; Vermont et al., 2016).

The coccolithophore Emiliania huxleyi is a globally distributed and abundant oceanic phytoplankton species whose blooms can cover thousands of square kilometers (Holligan et al., 1993). They are a key component in pelagic food webs contributing essential amino acids and fatty acids (FA) to the base of the food chain, which are crucial for supporting multiple cellular functions and growth in higher trophic level organisms. The collapse of E. huxleyi blooms has been linked to infection by double-stranded (ds) DNA viruses (EhVs) (Bratbak, Egge & Heldal, 1993; Brussaard et al., 1996; Wilson et al., 2002). Infection with EhV causes rapid physiological changes in E. huxleyi that divert host resources toward virus replication and assembly; e.g., decreased photochemical efficiency (Gilg et al., 2016) and altered metabolic pathways such as glycolysis, FA, and nucleotide biosynthesis (Evans, Pond & Wilson, 2009; Malitsky et al., 2016; Rosenwasser et al., 2014). Within three hours post inoculation with EhV, E. huxleyi cultures shift from producing polyunsaturated (PUFA) to monounsaturated (MUFA) and saturated (SFA) fatty acids (Floge, 2014). Additionally, viral infection can increase the uptake capacity of N and P by expressing viral genes which code for nutrient transporters not found in the host’s genome and increase access to diverse nutrient sources unavailable to uninfected cells (Monier et al., 2017; Monier et al., 2012; Wilson, Carr & Mann, 1996). High P and/or N resources are critical for optimal viral proliferation in phytoplankton hosts (Maat & Brussaard, 2016; Maat et al., 2016; Mojica & Brussaard, 2014; Monier et al., 2017), including E. huxleyi (Bratbak, Egge & Heldal, 1993; Martínez Martínez, 2006). At the scale of large oceanic E. huxleyi blooms it remains unclear if the sum of viral alterations enhances or diminishes the overall amount of C and essential nutrients that are passed to higher trophic levels.

Predation by heterotrophic and mixotrophic protists (microzooplankton) dominates grazing on phytoplankton in aquatic microbial food webs, and plays a key role in C cycling and nutrient regeneration (Calbet & Landry, 2004; Sherr & Sherr, 2002; Sherr & Sherr, 2009; Strom et al., 2001). The heterotrophic dinoflagellate Oxyrrhis marina has been shown to preferentially graze on EhV-infected E. huxleyi cells, relative to uninfected cells (Evans & Wilson, 2008). Although the mechanism driving this preferential grazing is unclear, Evans & Wilson (2008) proposed possible changes in prey size, motility, nutritional value, palatability, and chemical cues as potential causes. However, to the best of our knowledge, those results have not yet been reproduced in any independent studies. Also, one aspect that was not investigated is if and how preferential grazing on infected E. huxleyi might modify the transfer of C and essential nutrients through the food web. In this study we investigated the effect of viral infection on the growth and ingestion rates, cell volume, and FA composition of O. marina cells to better understand how the grazing behavior and physiology of microzooplankton is influenced by viral infection of abundant and important phytoplankton prey.

Materials and Methods

Culture maintenance

Clonal Emiliania huxleyi strain CCMP 374 (non-axenic, non-calcifying; 3-5µm) and non-axenic clonal O. marina strain CCMP 1795 were obtained from the Provasoli-Guillard National Center for Marine Algae and Microbiota (NCMA-Bigelow Laboratory, Boothbay, ME, USA). A non-axenic clonal Dunaliella sp. strain was sourced from the University of South Carolina. E. huxleyi, O. marina, and Dunaliella cultures were maintained at 16 °C under a light:dark cycle (14:10 h; 250 µmol photons m−2s −1). E. huxleyi and Dunaliella sp. cultures were kept in exponential growth phase by periodically transferring 10% (v/v) culture into fresh f/2-Si seawater medium (Guillard, 1975). Under these standard culture conditions E. huxleyi CCMP 374 cultures display near-synchronous division that starts approximately 1 h before the onset of the light period and last approximately 4 h (Gilg et al., 2016). O. marina stock cultures were fed weekly with fresh Dunaliella sp. cultures (5% (v/v)). Fresh EhV-86 (Wilson et al., 2002) lysates were obtained by inoculating E. huxleyi cultures in exponential growth phase. Once culture clearance was observed (typically 3–5 days post inoculation (p.i.)), cell debris was removed by filtration (0.45 µm PES filter) and the EhV-86 lysates were then stored at 4 °C in the dark for up to two weeks prior to being used in an experiment. The same EhV-86 lysate stock was used to determine infection dynamics and for grazing experiments 1–3 (see experimental details in the sections below). Cell and virus concentrations were measured using a FACScan flow cytometer (Beckton Dickinson, Franklin Lakes, NJ, USA), equipped with an air-cooled laser providing 50 mW at 488 nm with standard filter set-up, as previously described (Brussaard, 2004; Marie et al., 1999). Virus particles and bacteria cells were enumerated from 1 ml 0.5% glutaraldehyde-fixed (final concentration) samples that were frozen in liquid nitrogen and stored at −80 °C until further processing. The samples were thawed and stained with SYBR Green I prior to flow cytometry (FCM) enumeration. Virions and bacteria were discriminated based on green fluorescence and side scatter signals (Fig. S1A). Emiliania huxleyi and O. marina cells were visualized and enumerated together from fresh, non-fixed 1 ml samples by triggering the cytometer on red fluorescence. Cells were enumerated based on chlorophyll red auto-fluorescence and side scatter (Figs. S1B, S1B, S1D). Note that red fluorescence in fed O. marina cells derived from ingested prey (Figs. S1C, S1D). Starved O. marina cells only show residual red fluorescence signal (Fig. S1B).

Emiliania huxleyi virus infection dynamics

Fifty milliliter aliquots of exponentially growing E. huxleyi culture were inoculated at four EhV-86 to host ratios of 5:1, 20:1, 50:1, and 100:1, in triplicate. The inoculations both here and for experiments 1–5 (see below) were timed to coincide with the end of the cytokinesis to minimize the effect of division on the estimation of infection and grazing rates. Fresh f/2-Si media was added to each flask in order to achieve the same E. huxleyi cell concentration in all flasks. One milliliter aliquots were taken from each culture at 2, 4, 6, and 20 h p.i. for cell enumeration using flow cytometry (FCM). Cells were stained with the orange fluorescent lipid-specific dye N-(3-Triethylammoniumpropyl)-4-[4-(dibutylamino)styryl] pyridinium dibromide (FM 1-43; Invitrogen Co., Carlsbad, CA, USA) to allow discrimination between visibly infected and non-infected E. huxleyi cells (Martínez Martínez et al., 2011) (Fig. S2). Progression of the viral infection was quantified by tracking the percentage of visibly infected E. huxleyi over time.

Emiliania huxleyi C and N content

A 150 ml volume of E. huxleyi culture in exponential growth phase was divided in two equal volumes. One of the aliquots received 45 ml EhV-86 to achieve a 50:1 virus:host ratio; the second one received an equal volume of fresh f/2-Si medium to achieve similar cell concentration in both cultures. Emiliania huxleyi concentration and percentage of visibly infected cells in each flask was determined immediately after the addition of EhV-86 and f/2-Si medium and at 5 h p.i., and at 24 h p.i. Six 5 ml samples were taken from each flask immediately after the addition of EhV-86 and f/2-Si medium and at 24 h p.i. and were gravity filtered through a combusted glass fiber filter (Whatman GF/F; GE Healthcare Life Sciences, Pittsburg, PA) to collect particulate matter. The filtrates were then passed through fresh combusted GF/F filters to serve as C and N background controls (residual dissolved C and N retained by the filters). Filters were stored at −80 °C until analysis. Prior to testing, the filters were dried at 45 °C for 24 h before being placed in 9 × 10 mm Costech tin capsules using clean forceps and sample preparation block. Calibration standards were prepared from acetanilide. The samples, standards, and filter blanks were analyzed using a Costech ECS 4010 elemental analyzer (980 °C combustion).

Oxyrrhis marina specific growth and grazing rates

Five independent experiments (experiments 1–5) were performed. Oxyrrhis marina was not fed for 3 days prior to each experiment to ensure their feeding vacuoles were empty. FCM was employed to check for the absence of prey-derived chlorophyll red autofluorescence signal within O. marina vacuoles after the 3-day period. For experiments 1–4, stock non-axenic E. huxleyi cultures (∼1 × 106 cells ml−1) were split into two equal volumes. One of the flasks was inoculated with fresh EhV-86 lysates to achieve the virus:host ratios specified for each experiment in Table 1. Incubations were carried out for 6 h to allow sufficient viral infection levels (see results from virus infection dynamics below, Fig. 1). The second flask received f/2-Si media equal to the virus stock volume to match the dilution of cells. For experiment 5, E. huxleyi culture was infected as described previously and was split into two equal volumes at 6 h p.i. One aliquot was kept unfiltered and the other one was filtered through a 0.4 µm pore size polycarbonate filter. The filter pore size was tested prior to the experiment to ensure selective removal of E. huxleyi cells (<1% cells passed through). Between 78% and 100% bacteria and 75–95% EhVs were allowed through the 0.4 µm filter. Experiment 5 was conducted to determine if O. marina growth rate is augmented or supported by ingested bacteria, virions, and/or dissolved organic matter (DOM) within the infected cultures. Non-infected diet was not included in experiment 5.

Table 1 Details of experiments performed.

	Experimental conditions	Parameters measured (Om)	
	EhV: Eh ratio	% infected Eh cells 6 h p.i.	Eh:Om ratio	Initial Om (cells/ml)	Duration (days)	Growth rate	Grazing rate	Fatty acids	Cell vol	
Experiment 1	100:1	20	30:1	6000	0.25		x			
Experiment 2	100:1	20	100:1	4500	3	x	x	x		
Experiment 3	100:1	20	100:1	4000	7	x	o			
Experiment 4	50:1	36	100:1	6000	4	x	x		x	
Experiment 5	50:1	34	100:1	6000	4	x	x			
Notes.

Eh Emiliania huxleyi

Om Oxyrrhis marina

o indicates that the grazing rates were calculated using E. huxleyi k-values from non-grazing controls in experiment 2.

Figure 1 Infection progression of E. huxleyi.

Infection progression of E. huxleyi at four different virus:host ratios; 5:1(▴), 20:1 (♢), 50:1 (■), and 100:1 (●). Values are mean percentage (%) of cells visibly infected over time (hours) ± one standard deviation.

In all experiments, the flasks were incubated without shaking under the standard culture conditions indicated above. Equal volume aliquots of either infected (including 0.4 µm-filtered) or non-infected E. huxleyi cultures were fed to triplicate O. marina cultures. Both infected and non-infected food contained bacteria; additionally, the infected food contained free EhV particles. Additional aliquots of the E. huxleyi cultures (infected and non-infected) were maintained separately as non-grazing controls. It should be noted that non-grazing control cultures were not maintained during experiment 3, instead, average E. huxleyi growth rates from experiment 2 were used to normalize for E. huxleyi cell growth and lysis. Both these experiments employed the same EhV stock and virus:host ratio, and only differed in the length of time of the experiment (Table 1). We have shown in this study and elsewhere (Gilg et al., 2016; Vermont et al., 2016) that under comparable conditions, infection dynamics and virus production are highly reproducible. E. huxleyi and O. marina cell concentrations were monitored in each flask by FCM. Prey and predator cell concentrations were measured immediately after the initial feeding and every 30 min for the first 2 h and then every hour up to 6 h for experiment 1 (Fig. S3) and every 24 h for experiments 2–5. During experiments 2–5 O. marina cultures were fed fresh prey immediately after determining cell concentrations at the end of each 24 h incubation period for a total of 3–7 days (Table 1). The duration of our experiments are ecologically relevant and representative of high rates of viral infection during induced blooms of mixed assemblies of E. huxleyi (Castberg et al., 2001; Martínez Martínez et al., 2007). The additions of fresh prey cells each day were calculated to bring the prey:predator ratio to the same level as at the beginning of the experiment. Sterile f/2-Si medium was added, as needed, to the O. marina cultures to maintain comparable cell concentration between treatments. Additional experimental design information can be found in Table 1. Oxyrrhis marina specific growth and grazing rates were determined by the equations of Frost (1972) and used to calculate O. marina growth per E. huxleyi cell consumed.

Projected O. marina’s abundance

Our goal in these experiments was to maintain consistent concentration of O. marina and E. huxleyi throughout the experiment. At each time point the cultures were sampled and high concentration aliquots of fresh E. huxleyi cultures were added to replace the cells that were ingested or lost due to mortality. The concentration data with the daily dilution was used to calculate the total number of O. marina and E. huxleyi cells that were produced and consumed, respectively, over the duration of the experiment. Average growth and grazing rates of individual cultures from experiments 2–4 (n = 9) were combined to calculate the overall average ± 1 standard error (SE) growth and grazing rates of O. marina fed either infected or non-infected E. huxleyi cells. We postulated a starting population size of 6,000 O. marina cells and assumed E. huxleyi prey saturation and no mortality for O. marina over a 7-day period. We applied the overall O. marina’s average ± 1 SE growth rate over the 7-day period to calculate the cumulative population size supported by infected or non-infected E. huxleyi. We then used the calculated average population size of O. marina and its average grazing rates (±1 SE) (Frost, 1972) to calculate the total ingestion of E. huxleyi cells at each time point.

Oxyrrhis marina and E. huxleyi fatty acid (FA) analysis

The effect of feeding on virally infected or non-infected E. huxleyi on the FA composition of O. marina was investigated during experiment 2. For FA analysis, 5 ml aliquots were taken from non-infected E. huxleyi cultures and from cultures 6 h after inoculation with EhV-86 (in duplicate), as well as from O. marina cultures (in triplicate) before feeding them with E. huxleyi cells (Day 0) and after three days being fed E. huxleyi (Day 3). Samples were vacuum filtered through a combusted glass fiber filter (Whatman GF/F; GE Healthcare Life Sciences, Pittsburg, PA), and stored at −80 °C until analysis. FAs were converted to FA methyl esters (FAMEs) in a one-step extraction direct methanolysis process (Meier et al., 2006) following the procedures detailed in Jacobsen, Grahl-Nielsen & Magnesen (2012).FAMEs were analyzed on a gas chromatograph with mass spectrometric detector (Shimadzu GCMS-QP2010 Ultra; Shimadzu Scientific Instruments, Columbia, MD, USA). FAME samples were reconstituted in 200 µl of hexane and 1 µl was injected into the GC/MS injector which was kept at 250 °C. FAMEs were separated on a SGE BPX-70 column, in a helium mobile phase at a flow rate of 1.17 ml min−1. A Supelco 37 Component FAME Mix (47885-U; Supelco Analytical, Bellefonte, PA, USA) standard solution was used for instrument calibration. Individual FAMEs were identified via comparison to standard mixture peak retention times and fragmentation patterns using the NIST-library of compound mass spectra. FAME concentrations were calculated from peak area relative to that of a C19:0 internal standard that was added to each sample prior to extraction. FA type concentrations were converted to percentages of the combined total FA concentration.

Oxyrrhis marina cell volume

The cell size of O. marina was measured for starved cells (3 days) prior to the experiment and once a day for three days from each O. marina culture during the grazing experiment 4. The size of O. marina was measured on fixed cells. Common fixatives such as Lugol’s or glutaraldehyde alter cell volume (Menden-Deuer, Lessard & Satterberg, 2001). Alternatively, live cells can be immobilized by adding nickel sulfate (0.003% final concentration) or 70% ethanol, which appears to have no effect on cell shape and size (Menden-Deuer, Lessard & Satterberg, 2001). We chose to fix O. marina cells by transferring 50% (v/v) into 70% ethanol and storing at 4 °C for 30 min prior to analysis. Ethanol did not appear to alter cell size since our results were very similar to the measurements with added nickel sulfate from Menden-Deuer, Lessard & Satterberg (2001); however, no direct comparison of these two methods was carried out. Ten randomly selected individual O. marina cells from each aliquot were photographed on a hemocytometer. Width and length of the cells were measured using ImageJ (Schneider, Rasband & Eliceiri, 2012). Volume was calculated as a rotational ellipsoid; V= π6xd2xh (Edler, 1979; Menden-Deuer & Lessard, 2000). Differences in cell volume between treatments were evaluated using a standard t-test. Total C per O. marina cell was estimated based on the average cell volume using the equation log pg C cell−1 = −0.665 + log vol × 0.939 (Menden-Deuer & Lessard, 2000).

Statistical analyses

Temporal differences within the same diet treatment for E. huxleyi C and N content and for O. marina specific growth and ingestion rates, and cell volume were analyzed with a two-tailed, paired t-test, Alpha level 0.01. When comparing parameters between treatments and for FA composition, the differences were analyzed with a two-tailed, unpaired t-test assuming equal variance, Alpha level 0.01. P-values (P < 0.05) were significant and P < 0.01 were considered highly significant.

Results

Emiliania huxleyi virus infection dynamics

The percentage of visibly infected cells (as revealed by FCM) increased at higher virus:host inoculation ratios over a 20 h period. During this same period cell abundance did not change significantly in virally-infected cultures compared to non-infected cultures (Fig. S2C). The highest virus:host ratio (100:1) yielded at least 20% visibly infected E. huxleyi cells by 6 h p.i., and ∼57% by 20 h p.i. (Fig. 1); consequently we chose this ratio for experiments 1–3, which were carried out with the same EhV-86 stock and under the same environmental conditions employed to determine the infection dynamics (Fig. 1). The infection dynamics of E. huxleyi (CCMP374) and viral (EhV-86) production are highly consistent and reproducible when using the same host and virus strains and conditions, in particular when using the same virus lysate stock within a 4-week time frame (Gilg et al., 2016; Vermont et al., 2016). Infected cells begin to release virus progeny at around 4.5 h p.i. (Mackinder et al., 2009) and any cells not infected by the initial EhV inoculum can become infected during successive infection rounds by the new EhV progeny. At the high virus:host ratios used in this study, previous work has shown that 70%–100% of the E. huxleyi cells become infected by 24 h p.i. (Gilg et al., 2016; Vermont et al., 2016), even if not evident by FCM (Martínez Martínez et al., 2011). In experiments 4 and 5, we used fresh EhV-86 lysate stocks that yielded an apparent higher percentage of infected cells by 6 h p.i. than the EhV-86 stock used in the three preceding experiments. Consequently, we reduced the initial virus:host ratio to 50:1 virus:host ratio to achieve more comparable infection dynamics (∼36% and ∼34% of E. huxleyi cells were visibly infected by 6 h p.i. in experiments 4 and 5, respectively).

Emiliania huxleyi C and N content

Staining with the lipid-specific dye FM 1-43 showed ∼15% visibly-infected E. huxleyi cells by 5 h p.i. and over 70% visibly-infected cells by 24 h p.i. Over a 24 h incubation, both infected and non-infected cultures of E. huxleyi exhibited a slight but significant increase (P = 4.94 ×10−4 and P = 0.018, respectively) in C content of 9.27 ± 0.19 to 10.86 ± 0.43 pg C cell−1 (±SD) for infected cells and 8.94 ± 0.94 to 10.42 ± 0.26 pg C cell−1 for non-infected cells, respectively. Carbon content was not statistically different between treatments (P = 0.420 at t0 and P = 0.057 at t24) (Fig. 2A, Table S1). Emiliania huxleyi N content was not statistically different between samples at the beginning of the experiment (P = 0.989), but it increased significantly over the 24 h incubation; from 1.51 ± 0.08 to 1.89 ± 0.13 pg N cell−1 (±SD) (P = 6.36 ×10−4) for infected cells and 1.51 ± 0.05 to 2.09 ± 0.07 pg N cell−1 (P = 1.80 × 10−5) for non-infected cells, respectively. The N content of non-infected cells was significantly higher after the 24 h incubations (P = 0.008) (Fig. 2B).

Figure 2 E. huxleyi C and N concentration.

E. huxleyi C (A) and N (B) concentration (pg cell −1) at 0 and 24 hours p.i. Values are mean ± one standard deviation. Letters indicate statistical similarity. Same letters indicate no statistical difference between compared treatments and different letters denote significant statistical differences.

Oxyrrhis marina specific growth rate

The 6 h duration of experiment 1 was too short to measure O. marina growth rates. In the longer experiments (experiments 2–4), the growth rates of O. marina ranged from 0.28 to 0.43 day −1(average 0.35 ± 0.08 day−1 (±SD)) when fed non-infected prey and from 0.47 to 0.56 day −1(average 0.52 ± 0.05 day−1) when fed infected prey (Fig. 3A, Table S2). Specifically, O. marina specific growth rates were 30% (P = 0.002), 43.4% (P = 5.29 ×10−6), and 91.3% (P = 0.006) higher when fed infected E. huxleyi during experiments 2, 3, and 4, respectively (Fig. 3A). Based on the average growth rates, and assuming no loss term for O. marina cells, we calculated the abundance of O. marina cells after 7 days was 233% higher on a diet of infected cells than on non-infected E. huxleyi cells (Fig. 3D). In a follow up experiment (experiment 5) that measured the ingestion and growth rates of O. marina on infected E huxleyi and bacteria, the data showed that the growth rate of O. marina was significantly higher (P = 6.49 ×10−4) when fed infected, non-axenic E. huxleyi cells (average 0.38 ± 0.02 day−1 (±SD)) than when fed <0.4 µm filtrate from infected cultures (average 0.10 ± 0.05 day−1 (±SD)) (Fig. 4A, Table S3).

Figure 3 Growth and grazing rates.

Differentialgrowth and grazing rates of O. marina fed non-infected versus infected E. huxleyi. (A) O. marina growth rates (day −1). (B) O. marina grazing rates (Eh cells Om −1h−1 (Exp. 1) or Eh cells Om −1 day −1 (Exps. 2–4); see Table 1). (C) O. marina growth rate divided by grazing rate (O. marina divided per E. huxleyi consumed). Values mean ± one standard deviation (Experiments 2, 3, and 4) and standard deviation (Experiment 1). (D) Projected O. marina’s population size at each time point. E: Projected total consumption of E. huxleyi cells at each time point. Dashed lines are average values and shaded regions are one standard error from Experiments 2, 3, and 4. Asterisks indicate statistical significance: *** p < 0.001, ** p < 0.01, * p < 0.05.

Figure 4 Differential growth and grazing rates measured in experiment 5 for O. marina fed non-axenic infected E. huxleyi versus <0.4 µm filtrate of a non-axenic infected E. huxleyi culture.

(A) O. marina growth rates (day−1). (B) O. marina grazing of E. huxleyi cells (Eh cells Om−1 day −1). (C) O. marina grazing of bacteria cells (Bact cells Om−1 day−1). Values mean ± one standard deviation. nd denotes “none detected”. Asterisks indicate statistical significance: ***p < 0.001, **p < 0.01.

Oxyrrhis marina ingestion rates

During the initial 1.9 h in experiment 1, O. marina ingestions rates were not significantly different (P = 0.68) when fed infected (14.48 ± 0.18 Eh cells Om−1 h−1 (± SD)) versus non-infected cells (14.88 ± 1.57 Eh cells h−1). Between 1.9 (experiment 1) and 6 h (experiment 2) no additional ingestion was measurable (Fig. 3B, Fig. S3, Table S2). Initial pulse-feeding following a period of starvation is commonly observed in grazing experiments and may explain the equal number of prey ingested in experiment 1 (1.5 h) and 2 (6 h). The combined results from grazing experiments 2–4 yielded ingestion rates that were on average 35.4% lower (P = 0.001) for O. marina fed infected (39.74 ± 12.14 Eh cells Om−1 day−1) versus non-infected E. huxleyi cells (60.34 ± 10.13 Eh cells Om−1 day−1) (Fig. 3B, Table S3). The higher total number of ingested E. huxleyi cells, both infected and non-infected, measured in experiments 2 –4 versus experiment 1indicated that O. marina can saturate its feeding vacuoles within the first 1.5 h of feeding and then resumes feeding after 6 h, as prey cells were digested. Also, the relatively low standard deviation values between experiments 2–4 indicated daily ingestion rates were fairly constant over time. Normalizing O. marina growth rate to the number of cells ingested renders differences between diets (P = 1.60 ×10−5) even more striking (i.e., 86.30%, 238.62%, and 154.44% higher growth rate when fed infected E. huxleyi cells, for experiments 2, 3, and 4 respectively) (Fig. 3C, Table S2). The elevated growth rate in of O. marina in the infected cultures was driven largely by ingestion of E. huxleyi cells and not from ingested bacteria within the cultures (Fig. 4). A comparison of the growth rates of O. marina feeding on infected cultures showed that when the E. huxylei cells were removed, O. marina growth rates decreased precipitously from 0.38 d−1 (± 0.02 SD) to 0.1 d−1 (± 0.05) (Fig. 4A). Ingestion rates of O. marina on infected E. huxleyi cells were similar to experiments 2–4 (64.98 ± 2.72 Eh cells Om−1 day−1). However, in the <0.4 µm filtered treatment where approximately 99% of the E. huxleyi cells were removed, the ingestion of E. huxleyi decreased by 95%, (3.5 ± 4.04 Eh cells Om−1 day−1) on day 1 and was undetectable on the subsequent 3 days (Fig. 4B, Table S3). Conversely, the ingestion of bacteria cells (Fig. 4C) in the flasks with high abundance of E. huxylei cells showed low ingestion rates of bacteria on day 1(262 ± 64.51 bact cells Om−1 day−1) and no detectable ingestion on the subsequent days of the experiment. However, when the E huxleyi cells were removed, bacteria ingestion rates increased from 767 ± 130.35 bact cells Om−1 day−1 on day 1 to 3132 ± 455.02 bact cells Om−1 day−1 on day 4. Grazing of bacteria cells was statistically different (P = 3.17 ×10−3) between diets on day 1 (Fig. 4C, Table S3).

Combining the higher growth rate of O. marina (i.e., higher end abundance, Fig. 3D) and the average ingestion rates (Fig. 3B), we estimated that the total consumption of virus-infected E. huxleyi cells would exceed that of non-infected cells after 4–5 days and would be on average 63.2% higher for virus-infected E. huxleyi over a 7-day period (Fig. 3E).

Oxyrrhis marina and E. huxleyi fatty acid analysis

Fatty acid (FA) profiles were similar between infected and non-infected cultures of E. huxleyi (note the low SD values between the concentrations of the most abundant FAs in the duplicate non-infected and duplicate infected E. huxleyi cultures). (Table 2). Similarly, minor differences in the FA profile were observed in O. marina that had consumed infected versus non-infected cells. The cultures containing O. marina fed infected E. huxleyi contained slightly higher proportions of C17:0 and 2-fold higher proportions of C20:2 (Table 2).

Table 2 Percentage (%) of individual fatty acids to total FA concentration of cultures in which: (i) O. marina was depleted of prey at the start of the experiments (Day 0 Om); (ii) the E. huxleyi cultures fed to O. marina; (iii) after three days fed non-infected E. huxleyi (Om + Eh non-inf); and (iv) after 3 days fed infected E. huxleyi (Om +Eh inf).

Values are mean ± one standard deviation, n = 4, n = 4, n = 3 and n = 3, respectively. Note the values for E. huxleyi are the average of duplicate non-infected and duplicate infected cultures.

FA Class	Day 0 Om	E. huxleyi	Day 3 Om + Eh non-inf	Day 3 Om + Eh inf	
SFA	C14:0	0.2 ± 0.2	0.2 ± 0.1	0.4 ± 0.1	1.2 ± 1.3	
C15:0	0.4 ± 0.3	0.3 ± 0.2	0.5 ± 0.0	0.8 ± 0.4	
C16:0	30.9 ± 4.5	27.6 ± 3.6	29.4 ± 7.6	24.6 ± 2.6	
C17:0	2.7 ± 0.1	2.7 ± 0.7	2.7 ± 0.3	3.2 ± 0.1c,d	
C18:0	36.2 ± 11.1	60.9 ± 7.6a	46.9 ± 6.4	56.4 ± 7.1c	
C20:0	1.1 ± 0.1	1.3 ± 0.3	1.5 ± 0.8	1.2 ± 0.4	
C22:0	0.8 ± 0.2	0.7 ± 0.6	3.3 ± 2.3	1.8 ± 0.9	
C24:0	0.7 ± 0.5	1.1 ± 0.1	0.6 ± 0.4	0.4 ± 0.3	
MUFA	C16:1	0.8 ± 1.6	0.0	0.0	0.0	
C18:1(n-9cis)	5.0 ± 4.0	0.0a	1.6 ± 1.4b	1.6 ± 1.6c	
C18:1(n-9trans)	4.6 ± 3.3	0.0a	1.7 ± 1.5	1.3 ± 1.2	
C22:1	1.1 ± 0.9	0.1 ± 0.1	0.0	0.0	
PUFA	C18:2	2.7 ± 1.4	0.0	0.0 b	0.2 ± 0.3c	
C20:2	2.9 ± 2.3	5.0 ± 3.7	7.1 ± 1.9b	3.2 ± 1.3c,d	
C20:5 (n–3)	1.2 ± 1.2	0.1 ± 0.1	0.2 ± 0.3	0.3 ± 0.3	
C22:6 (n-3)	8.6 ± 7.7	0.0	4.0 ± 4.1	3.9 ± 3.3	
	∑ SFA	73 ± 15	95 ± 4a	86 ± 9	89 ± 6	
	∑ MUFA	12 ± 8	0.1 ± 0.1a	3 ± 3	3 ± 3	
	∑ PUFA	15 ± 8	5 ± 4	11 ± 6	8 ± 3	
Notes.

Significant differences in the proportions of individual compounds are shown as: a Day 0 Om vs. E.huxleyi

b Day 0 Om vs. Day 3 Om + Eh non-inf

c Day 0 Om vs. Day 3 Om + Eh inf

d Day 0 Om vs. Day 3 Om + Eh non-inf vs. Day 0 Om vs. Day 3 Om + Eh inf

Oxyrrhis marina cell volume

The average volume of O. marina cells was slightly larger (∼10%) when fed a diet of infected E. huxleyi prey (5,226 ± 1,267 µm 3(±SD)) than when fed non-infected E. huxleyi (4,706 ± 1,259 µm3). However, the difference was not statistically significant (P = 0.21) (Table 3, Fig. S4). While the SD were large, possibly due to the relatively low number of cells that were measured for each time point and treatment, they were similar for both diet treatments suggesting similar high volume variations between individuals in both treatments.

Table 3 Cell volumes (µm3) of O. marina fed non-infected and infected E. huxleyi over three days during experiment 4.

Treatment	Day	Replicate	Volume (µm3)	
Om prey-depl	0		5586 ± 917	
Om + Eh non-inf	1	A	4723 ± 1535	
B	5759 ± 1123	
C	5696 ± 1842	
2	A	4107 ± 1689	
B	3801 ± 660	
C	5602 ± 1045	
3	A	4675 ± 1141	
B	4038 ± 1319	
C	3949 ± 977	
Om + Eh inf	1	A	5004 ± 1245	
B	4286 ± 1053	
C	4829 ± 1435	
2	A	6105 ± 462	
B	5561 ± 1483	
C	6977 ± 1371	
3	A	4267 ± 1218	
B	4478 ± 1184	
C	5527 ± 1956	
Notes.

Om Oxyrrhis marina

prey-depl prey-depleted, i.e., not fed for three days

Eh non-inf Emiliania huxleyi non-infected

Eh inf Emiliania huxleyi infected with EhV-86

Values are mean ± one standard deviation.

Discussion

The results presented here show compelling evidence that virus-infected E. huxleyi fuels higher growth rates in the heterotrophic dinoflagellate O. marina. The data shows that the higher growth rates of O. marina resulted from ingesting fewer E. huxleyi cells than non-infected cells suggesting higher nutritional value of the infected E. huxleyi cells. Experiment 5 shows that the enhanced growth rates were not due to the ingestion of bacteria, virions, or DOM within the infected cultures. Furthermore, the fast growing O. marina, showed similar FA profile and cell size when compared to O. marina reared on non-infected algae suggesting that the quality of the dinoflagellates to higher trophic levels is unchanged. Consequently, the higher growth efficiencies of O. marina feeding on virally infected E. huxleyi cells suggest that viral infection of E. huxleyi increases the production of microzooplankton O. marina. These results suggest a shift in the “viral shunt” paradigm in which the flow of organic matter to higher trophic levels is enhanced by viral infection of algae rather than being short-circuited.

Oxyrrhris marina feed and grow on a wide range of protist prey types, as well as bacteria (Jeong et al., 2008) and DOM (Lowe et al., 2011). However, some prey enhance growth rates more than others (Montagnes et al., 2011). For example, virus-infected E. huxleyi cells supported higher O. marina growth than non-infected E. huxleyi cells, despite lower ingestion rates, suggesting higher nutritional value or higher assimilation efficiency of infected prey cells. It is worth noting that O. marina ingestion rates on virally infected E. huxleyi cultures might have been overestimated. A reduction in prey abundance due to viral lysis over each 24 h interval, prior to fresh-prey replenishment, during the experiments might have led to temporarily reduced grazer-prey encounter and ingestion rates. Under such scenario O. marina’s growth per ingested infected cell would have been even larger than we estimated, adding further significance to our results. Consequently, our study would represent a conservative estimate of C transfer efficiency. This study also shows that O. marina only ingested bacteria when E. huxleyi cells were not available (or in very low abundance) and that ingestion of bacteria cells (and possibly EhV particles and/or DOM) alone does not support high growth rates compared to E. huxleyi cells even when bacteria are in very high concentrations (107 cells ml−1).

Based on our measurements, the mechanisms underlying the lower ingestion rates of virally infected E. huxleyi cells and higher growth efficiency remain unknown. Calcification reduces digestion efficiency and predator growth (Harvey, Bidle & Johnson, 2015). In the environment, E. huxleyi cells lose their liths during an active viral infection (Balch et al., 1993; Brussaard et al., 1996; Frada et al., 2008; Holligan et al., 1983; Jacquet et al., 2002; Trainic et al., 2018). In this study we chose a non-calcifying E. huxleyi strain to uncouple the effects of calcification and prey size, on feeding and growth rates. Furthermore, we found no differences between C content in infected and non-infected E. huxleyi cells. In contrast, N content was slightly higher in non-infected cells. Nitrogen depletion in some prey cells causes O. marina to cease grazing, although the mechanisms remains unknown (Flynn, Davidson & Cunningham, 1996; Martel, 2009). However, C:N ratios in all of our E. huxleyi cultures (virus-infected or non-infected) were lower than 6.6, indicative that N was replete (Davidson et al., 2005; Flynn, Davidson & Leftley, 1994).

Large dsDNA viruses of eukaryotic algae, such as EhVs, have a high demand of C, N and P for the production of lipids, proteins, and nucleotides to support typical high burst sizes. Viral infection can modulate host metabolic pathways and nutrient uptake to fulfill the metabolic requirements of viral production (Malitsky et al., 2016; Monier et al., 2017; Monier et al., 2012; Rosenwasser et al., 2014; Wilson, Carr & Mann, 1996). The production of intermediary biomolecules and changes in E. huxleyi’s lipidome induced by infection with EhV-86 (Evans et al., 2007; Evans et al., 2006; Evans, Pond & Wilson, 2009; Malitsky et al., 2016; Rosenwasser et al., 2014; Suzuki & Suzuki, 2006) potentially increase the nutritional value of infected cells. During EhV infection, changes in biosynthesis pathways result in the production of more highly saturated FAs (Evans, Pond & Wilson, 2009; Floge, 2014; Malitsky et al., 2016) and the enhanced production of sphingolipids (Pagarete et al., 2009; Rosenwasser et al., 2014). It should be noted that the majority of these virus-induced alterations in lipid composition have been detected after prolonged infection (>24 h) of E. huxleyi cultures. At the relatively coarse level of detail in lipid profile carried out in the present study, only minor differences in FA composition were observed between non-infected E. huxleyi cultures and cultures that had been infected for 6 h (Table 2). This suggests that differences in FA composition between recently-infected and non-infected E. huxleyi, were not responsible for the differences in growth rates of O. marina. However, it is possible that the relatively small sample volume collected for FA analysis of E. huxleyi cells limited the resolution and detection of differences in FA between infected and non-infected E. huxleyi cells (Evans, Pond & Wilson, 2009; Floge, 2014; Malitsky et al., 2016). Alternatively the 6 h time frame used in this study may be too short to measure significant changes in the FA profiles of infected cells. Alterations in lipid profile between infected and non-infected cells that were not apparent in our analysis (Hunter et al., 2015), may contribute to the higher growth efficiencies of O. marina fed virally infected prey cells.

An additional factor that influences the nutritional value of the phytoplankton prey is P concentration. In addition to providing a much needed resource for viral replication, P-rich phytoplankton cells increase grazing efficiency and secondary production in cladocerans (Elser, Hayakawa & Urabe, 2001; Sterner, 1993; Urabe & Sterner, 1996; Urabe & Watanabe, 1992). Low P availability reduces viral replication in E. huxleyi (Bratbak, Egge & Heldal, 1993) and other eukaryotic algae (Maat et al., 2016; Wilson, Carr & Mann, 1996), possibly by limiting the production of nucleic acids. It has been hypothesized that virally encoded putative phosphate transporters increase accumulation of P in host cells (Monier et al., 2012; Wilson, Carr & Mann, 1996). While to the best of our knowledge this has not been tested during the infection of E. huxleyi cells, most available EhV isolates, including EhV-86, carry an E. huxleyi- homolog putative phosphate repressible phosphate permease (PPRPP) gene (Martínez Martínez, 2006; Nissimov et al., 2011; Nissimov et al., 2012; Wilson et al., 2005), which we hypothesize led to higher P uptake in virally infected cells in our experiments. Additionally, the stoichiometric “light:nutrient hypothesis” poses that low supply of light relative to P yields more P-rich producers (i.e., low tissue C:P ratios) (Sterner et al., 1997); possibly due to the algae allocating high levels of P to light-harvesting cellular machinery and storing excess P intracellularly (Hessen, Færøvig & Andersen, 2002). In our study, E. huxleyi cells were grown in P-rich f/2-Si culture medium and both EhV-infected and non-infected cultures were kept under the same light conditions. In addition to the role of the PPRPP gene in P uptake, we hypothesize that virus-induced reduction in E. huxleyi’s photochemical efficiency from the early stages of EhV infection (Gilg et al., 2016) might also induce an increased P uptake and intracellular accumulation. While a reduction in photochemical efficiency might translate into lower C fixation rates, our results show that C content was not affected in infected compared to non-infected cells. Phosphorus content in infected and non-infected E. huxleyi cells and its impact on grazing warrants investigation in future studies.

E. huxleyi is an important food source at the base of the food chain and grazing pressure influences population and bloom dynamics (Fileman, Cummings & Llewellyn, 2002; Olson & Strom, 2002). Virus-induced mortality also plays a prominent role in bloom demise (Bratbak, Egge & Heldal, 1993; Brussaard et al., 1996; Castberg et al., 2001; Lehahn et al., 2014; Martínez Martínez et al., 2007) and diverts organic C away from upper trophic levels to the dissolved phase, which fuels the microbial loop—“viral shunt” (Wilhelm & Suttle, 1999). Our results suggest that viral infection also boosts microzooplankton production. High rates of viral infection can last from a few days (as in this study) to a few weeks during a natural E. huxleyi bloom progression (Brussaard et al., 1996; Castberg et al., 2001; Martínez Martínez et al., 2007), which could result in large differences in C flow through the food web. Extrapolating the results in our study, the enhanced growth rates of microzooplankton populations that feed on virally infected phytoplankton cells would lead to more organic C available for higher trophic levels. Thus, contrary to the idea that viral infection leads only to the production of dissolved organic matter (Wilhelm & Suttle, 1999), viral infections at the base of the food chain may augment the flow of C to higher trophic levels as well as toward the microbial loop. To the best of our knowledge, the specific functional response of copepods ingestion of O. marina fed infected and non-infected E. huxleyi has yet to be investigated; however, the nutrition and reproduction rates are enhanced in copepods fed O. marina (grown on other phytoplankton diets) compared to copepods that feed directly on small phytoplankton cells (Broglio et al., 2003; Chu, Lund & Podbesek, 2008; Parrish, French & Whiticar, 2012; Veloza, Chu & Tang, 2006). Phytoplankton are considered the primary producers of essential FA long chain n-3 (LCn-3) PUFAs; however, heterotrophic dinoflagellates such as O. marina are also able to produce sterols and essential FAs (e.g., EPA (C20:5 n-3) and DHA (C22:6 n-3)) from lipid precursors (Chu, Lund & Podbesek, 2008; Klein Breteler et al., 1999; Lund et al., 2008; Veloza, Chu & Tang, 2006), which emphasizes the important role of certain microzooplankton groups in trophic upgrading and C transfer and highlights the need for a better quantitative understanding of the factors that influence microzooplankton grazing behavior and secondary production rates. Incorporating quantitative data for viral lysis and the effect of viral infection in grazing behavior and transfer efficiency into ecosystem models is essential for accurate budgeting of C flow throughout the food web in the global marine ecosystem. As a cautionary reminder, when interpreting these results it is important to note that O. marina is not typically found in open waters (Yang, Jeong & Montagnes, 2011) and is not likely to be a common natural predator of E. huxleyi cells. However O. marina is frequently used as a model predator in laboratory-based experiments because of its morphological similarity to a wide variety of heterotrophic and mixotrophic dinoflagellates and its plasticity in feeding behavior allow it to represent a broad range of marine dominant microzooplankton (Lowe et al., 2011; Roberts et al., 2011). Furthermore, several studies have shown that O. marina responds to various experimental stimuli in similar ways to that of other microzooplankton taxa (Strom et al., 2003a; Strom et al., 2003b; Tillmann, 2004). E. huxleyi’s true protozoan predators in nature have yet to be precisely identified ( Wolfe, 2000).

A final consideration is that the lower ingestion rates of O. marina on E. huxleyi (strain CCMP374) cells infected with coccolithovirus EhV-86, compared to non-infected cells, are in contrast with an earlier study that used the same virus strain but a different E. huxleyi strain (CCMP 1516—non-calcifying) (Evans & Wilson, 2008). Strain-specific differences in the ingestion and clearance rates of O. marina feeding on E. huxleyi (Harvey, Bidle & Johnson, 2015) might have played a role in our findings. However, in light of our findings, the results from Evans & Wilson (2008) need to be revisited and revalidated and future studies should include multiple strains within a species (predator, prey, and/or virus) to test differences driven by intraspecific diversity. Importantly, future research is needed that focuses on a range of abundant and ecologically meaningful predator–prey-virus systems.

Conclusions

Viruses cause biochemical alterations to their E. huxleyi host cells to facilitate viral assembly (Gilg et al., 2016; Malitsky et al., 2016; Rosenwasser et al., 2014; Suzuki & Suzuki, 2006). The data presented in this study show that changes in E. huxleyi as a result of viral infection cause higher growth efficiency and an increase in heterotrophic protist production. Despite the faster growth rates, we found no major difference in cell size, total FA content or FA profile of O. marina maintained on a diet of virally infected E. huxleyi cells during 3 days as compared with O. marina individuals reared on non-infected cells for the same period of time. Combined, these results suggest that during viral infection of E. huxleyi, the flow of C to higher trophic levels increases. Thus, in addition to the “viral shunt” hypothesis, these results suggest that virally infected E. huxleyi cells may also shunt more C to higher trophic levels. In order to gain a more comprehensive understanding of ocean ecosystem webs it is essential that we get quantitative knowledge of the relative magnitude of each pathway. The significance of our work is that, given the global scale and rapid dynamics of viral infections in the ocean, infection of primary producers is likely to be one of the compounding factors that influences the qualitative and quantitative flow of C in oceanic systems and determines overall efficiency of transfer to higher trophic levels.

Supplemental Information

Supplemental Information 1 Supplementary Tables and Figures

Click here for additional data file.

Supplemental Information 2 Complete raw dataset

Click here for additional data file.

We acknowledge Bigelow Analytical Services for the analysis of the fatty acid and particulate carbon and nitrogen content.

Additional Information and Declarations

Competing Interests

Author Contributions

Data Availability

The authors declare there are no competing interests.

Andrew G. Goode conceived and designed the experiments, performed the experiments, analyzed the data, prepared figures and/or tables, authored or reviewed drafts of the paper, approved the final draft.

David M. Fields conceived and designed the experiments, performed the experiments, analyzed the data, contributed reagents/materials/analysis tools, authored or reviewed drafts of the paper, approved the final draft.

Stephen D. Archer performed the experiments, analyzed the data, prepared figures and/or tables, authored or reviewed drafts of the paper, approved the final draft.

Joaquín Martínez Martínez conceived and designed the experiments, performed the experiments, analyzed the data, contributed reagents/materials/analysis tools, prepared figures and/or tables, authored or reviewed drafts of the paper, approved the final draft.

The following information was supplied regarding data availability:

Raw data is provided in the Supplemental Information.

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
