# Peer review of "Physiological responses of Oxyrrhis marina to a diet of virally infected Emiliania huxleyi"

_PeerJ, doi:10.7717/peerj.6722_

## Round 0.1 · original submission · Major Revisions

Please note that all three reviewers felt this was an important and well-written manuscript, but they also had major concerns with the analyses and interpretations of the data. In addition, they suggested additional experiments that should be performed to serve as controls and help validate the conclusions.

Reviewer 1 ·

Basic reporting

Article well written and very interesting, but the results are ambiguous requiring additional experiments to validate the interpretations.
Literature, figures and tables are well presented.

Experimental design

Study requiring additional experiments to validate the interpretations as I elaborate below.

Validity of the findings

Please see my comment in the ' General comments for the author' section

Additional comments

In this study entitled ‘Physiological responses of Oxyrrhis marina to a diet of virally infected Emiliania huxleyi’ Goode and coauthors assessed the prey-predator interplay between the important, cosmopolitan bloom-forming phytoplankter, E. huxleyi, and O. marina a heterotrophic dinoflagellate frequently used as model microzooplankter in laboratorial assays. The results of the study performed in vitro using culturing and cell biology/biochemical approaches broadly showed, according to the authors, that O. marina displays higher grazing rates of non-infected E. huxleyi (Ehux) cells relative to infected cells, but contrastingly O. marina display higher growth rates when fed on infected cells. The mechanisms underlying these experimental observations are unknown, however the results are interesting and relevant leading the authors to suggest that virally infected Ehux can support larger growing populations of single celled heterotrophs and enhance carbon transfer to higher trophic levels.
From my view point this study highlights a very interesting phenomenon (although a previous study by Evans and Wilson have shown contrasting results - the authors acknowledge this fact) and clearly opens a new avenue of research. The study has various limitations when extrapolating to processes that might happen in natural bloom conditions. For example: 1) O. marina is not a common grazer during Ehux blooms (see Olson and Strom 2002) and therefore a more realistic response to natural predators remains unknown; 2) The Ehux strain (CCMP374) used is non-calcified. Yet, during bloom most cells appear to be calcified even during the bloom-demise phase and although noncalcified cells are produced probably as a result of viral infection these are minor in field populations (see Frada et al. 2012). A calcified strain should have clearly been used by the authors. Moreover, technically it does not seem to me that only 10 cells are sufficient to estimate significantly the average volume size of cells of a microbial population (see the large standard deviations obtained - How O. marina kept identical cell volumes while growing faster in the presence of infected cells? And it is not clear why infected Ehux cells were not included in the fatty acid (FA) analysis and how the authors distinguished between O. marina FA from the FA of prey FA in the food vacuoles during feeding conditions (where they starved prior to analysis?). Nonetheless, I consider that these represent some details that could either be mitigated by the authors here (some of them where discussed) or explored and developed in subsequent studies. However, what concerns me the most is the main result of the study, i.e. O. marina consumes less but faster in the presence of infected cells. As the authors suggested, this may indeed result from a higher nutritional value of infected cells or higher assimilation efficiencies, but it may have also resulted from additional processes that should have been considered by the authors. Specifically, O. marina can survive through the uptake of dissolved organic molecules in the laboratory and may be able to use this mechanism in saprobic environments (see Lowe et al. 2011) and can also feed on bacteria (Jeong et al. 2008). During a current lab or field Ehux infection, there are expected buildups of dissolved organic matter due to viral-mediated cell lysis, bacteria (cultures are not axenic) and off course virions in the medium (which can also be consumed by microzooplankton (see Gonzales and Suttle 1993), and possibly also by O marina as well). Thus, it is possible that the apparent reduced grazing on infected cells resulted in a shift in grazing preferences of O. marina to other highly abundant sources which could have supplemented enhanced growth rates of the grazer relative to cultures solely supplemented with non-infected cells. If this is the case the results and subsequent interpretation of the data will considerably change namely by emphasizing the importance of the viral shunt. Thus, I feel that an additional experiment testing this scenario assessing for example O. marina performances in conditioning medium from infections should be requested to the authors prior to publication.


A few more details:
Line 102: why O. marina was not maintained with Ehux 374 pior to the experiments? Please clarify.

line 43: Incorrect sentence. Modelling work – see Weitz et al. 2014 ISMEJ - show that viral infections led to an increase in productivity. Or are you referring productivity of the specific (target) host cells?

line 75: Full stop missing after “Martínez Martínez 2006)”.

Line 216: Please remove “and” between images were … measured.

Reviewer 2 ·

Basic reporting

Introduction
The introduction provides a good background to the study context, however there are a few references missing. For instance, on line 78 – 80, where microzooplankton contribution to grazing is mentioned, there is no reference to the Meta-analyses conducted by Calbet and Landry, 2004 or Schomker et al., 2013 that demonstrated that microzooplankton consume 62 – 67% of daily primary production.
Line 43: There has been some recent papers which have shown that viral lysis can lead to enhancement of primary production by releasing limiting nutrients. See (Shelford & Suttle, 2018, Biogeosci, 15, 809-819, Poorvin et al., 2004, Limno. Oceanogr, 49, 1734-1741, and Shelford et al., 2012, Aquat. Microb. Ecol. 66, 41-66).

Figures
Figure 1; Fig 1 should have a legend for each of the shapes and virus: host ratios. The data from this figure contrasts with line 244 – 246 where it is said after 24 hours close to 100% of E. huxleyi cells were infected, whereas this figure shows <60% were at 20h p.i.
Figure 2; On Fig 2b the letter c is above the third bar, it is not explained what this means, or if it should just be a b.
Figure 3; Fig 3d and e show the predicted model of growth and ingestion on E. huxleyi. The concentrations after 5 days for E. huxleyi ingestion (~3 x 106 (again assumed cells mL-1)) suggests an almost endless supply of E. huxleyi. It is also not clear whether the results are cumulative or per time point as O. marina abundance increases. This should be made clear in the text or figure legend.

Results
Line 273; it is not immediately clear from the Figure that the different experiments represent different end points of the experiment (if this is correct). I would suggest the authors add a comment to this in the figure text.
Line 291 – 292: In table 2 there is no data shown on recently infected E. huxleyi populations to back up this statement as the results have been combined.

Experimental design

Methods
Line 94 – it is said that the cultures are non-axenic, however there is no mention throughout the manuscript on bacterial abundances throughout experiments. This has a number of important implications, particularly as O. marina is able to feed on heterotrophic bacteria (see Jeong et al., 2008., Euk. Microb. 55,4, 271-288). This could affect the growth rates of O. marina if they are ingesting both prey types. Particularly as the release of DOM from lysed E. huxleyi cells could stimulate bacterial production.
On line 109-110 the method for cell and virus concentrations is reported, however the method for counting viral particles is not specifically reported (typically through SYBR-green or another stain, followed by heating and enumeration on a flow cytometer).
I think it would be useful, if O. marina was counted by flow cytometry to see an example plot in the supplementary information. From literature, and personal experience, it can be difficult to gain accurate numbers of dinoflagellates without using probes (e.g. acidotrophic stains).
It is not mentioned throughout the method section the volume of the experimental manipulations, rather just cell concentrations are reported. It would be useful to know this information, particularly to see if the volume removed from sampling is significantly reducing the volume in the flasks used.
Line 122 – 123; the authors did not specify where they used a 50:1 virus: host ratio for experiment 4, whereas all others used the 100: 1. This would complicate comparisons between experiment 4 and 1, 2 and 3. In a similar manner, E. huxleyi : O. marina ratios were different in experiment 1 which would affect the grazing rates measured. If there was a particular reason for this, it needs to be made clearer to the reader.
Line 125 – 126; the sentence is confusing with the use of the word “immediately”, this implies that there was a T0 sample taken, however this does not seem to be the case, and samples were only take 5h and 24h p.i. If T0 samples were not taken, it should be specifically mentioned in the text that the concentrations were assumed based on volume.
Line 177; the population size of O. marina is assumed 6,000 cells mL-1, but it is not defined here.
Line 177 – 178; the initial conditions used for predictive modelling of population changes uses a number of assumptions, which are not described. Firstly, the use of 6,000 O. marina cells (assumed as mL-1) as a starting density is very high, given that within field populations during E. huxleyi blooms (See Fileman et al., 2002) the maximum observed dinoflagellate abundance is 57 cells mL-1, and maximum within Olson & Strom, 2002 was ~200 cells mL-1. Secondly, the authors assume prey saturation, but provide no value for this. These conditions would need to be better defined if this is to be used within the manuscript.
On line 212 the authors report they decided to use the fixative 70% ethanol, and the results were similar to nickel sulfate in Menden-Deuer et al. (2001). Unfortunately, no comparison of methodologies was carried out, but it could be useful for the authors to compare with published values of O. marina sizes, to help prove their method.
Line 216, a word is missing between “Images were ( ) and ….”
It is not specifically mentioned if the authors included a monoculture of O. marina as a control population for any of the experiments. In addition, a control of O. marina with EhV lysate, just to be sure there is no impact of the heterotrophic dinoflagellate on the viral populations.

Validity of the findings

Results
Line 244 – 246: The authors mention that close to 100% of E. huxleyi cells were infected, even if not evident from FCM. It is not clear how this is assumed? An erroneous assumption here would be that all viruses produced are infectious, as there has been no evidence to suggest this (to my knowledge).

Discussion
Line 307 – 309: The fact there were no measures of bacterial abundance within the experiments means the authors cannot discredit the impact this may have had on O. marina growth rates. It should be specifically mentioned here.
Line 315 – 317: Did the authors measure a viral lysis rate from control E: huxleyi and EhV controls? This could be removed from the measured results to estimate the grazing impact of O. marina. This loss of E. huxleyi could also lead to enhanced DOM and bacterial production, as mentioned above.
Line 323 – 324: Although E. huxleyi do lose their coccoliths during infection (as seen in Frada et al., 2008) this was due to the shift from a diploid to haploid life style (“Cheshire Cat hypothesis”).
Line 337 – 338: Reference missing here of Hunter et al., 2015. Front. Mar. Sci. 2, 81 regarding lipidome changes during E. huxleyi infection.
Line 341 – 342: Same comment as above regarding sphingolipids.
Line 345 – 346: This conclusion is not properly shown in the results, especially given that they appear to be combined in Table 2.
Line 353 – The paper above by Hunter et al., 2015 displays changes in lipids during infection of E. huxleyi, therefore these could contribute towards the growth rate differences, however this paper isn’t referenced.
Line 378 – 379: Population and bloom dynamics of E. huxleyi by microzooplankton has been shown in a paper by Mayers et al., 2018. Prog. Oceanogr. doi: 10.1016/j.pocean.2018.02.024, in bloom and non-bloom conditions within the field.
Line 400 – 401: Trophic upgrading was not shown for all microzooplankton (currently only dinoflagellates). No trophic upgrading was displayed for a ciliate feeding on Dunaliella (Klein Breteler et al., Mar. Eco. Prog. Ser. 2004. 274, 199-208.

Additional comments

This paper by Goode et al, provides evidence for the impact of grazing by the dinoflagellate Oxyrrhis marina on virally infected and non-infected populations of the coccolithophore Emiliania huxleyi. This is an important study, as it focuses on the dual nature of different mortality pathways, which are often considered in isolation during laboratory experiments. Their results that O. marina growth rates are enhanced when feeding on infected cells is interesting, giving that this will affect E. huxleyi population dynamics within bloom scenarios.
The English is in general, well written throughout the manuscript, and ideas are conveyed well to the reader.
There are however a few methodological questions which are raised, primarily because bacterial abundance was not quantified throughout the experiment. The lysis of infected cells could lead to enhanced DOM release and enhanced bacterial growth, which could be leading to the higher growth rates observed by O. marina. However, no mention to this is present throughout the paper, in the methods, results or discussion.
In general, the methods section was a little difficult to understand on a first read, and they could perhaps be better explained to guide the reader through the procedures used.
In the methods section it is not described what piece of equipment the experiments are conducted in. The authors also do not describe if the incubation apparatus is maintained on a shaker, or is occasionally mixed to reduce settling of the material within the experiment.
With the fatty acid analysis conducted, the samples collected after 3 days would be a mixture of E. huxleyi and O. marina, as no size fractionation was conducted of the culture. This makes it difficult to compare with the initial analysis and to draw conclusions on how the fatty acid profiles differ. It should be specifically mentioned in the results and discussions section that we could not draw a conclusion from which portion of the community the results are.
Greater controls are needed within this experiment, for instance, having O. marina cultured with EhV filtrate. This would help eliminate any questions regarding whether O. marina is affected by viruses, or vice versa. This could be due to chemical mechanisms, or whether the feeding currents generated by O. marina could lead to virus ingestion or movement, which could influence predator-prey dynamics.

Reviewer 3 ·

Basic reporting

Yes on all accounts.
Providing the raw data was extremely helpful in better assessing this manuscript.

Experimental design

Clear and unambiguous, professional English used throughout.
Yes

Literature references, sufficient field background/context provided.
Yes

Professional article structure, figs, tables. Raw data shared.
Yes. Having the raw data was very helpful to further assess their results and conclusions!

Figures should be relevant to the content of the article, of sufficient resolution, and appropriately described and labeled.
Yes

All appropriate raw data has been made available in accordance with our Data Sharing policy.
Yes

Self-contained with relevant results to hypotheses.
Yes

Validity of the findings

The overall rationale to conduct these experiments is interesting and valuable to investigate. These types of experiments should give important insights into the complex interplay of virus-host-grazer interactions. However, the experimental set up and analyses, overall, need some major improvement or justification to convince me that virally-infected E. hux are actually resulting in higher growth rates in O. marina. In particular, they need to address the possible contribution of bacterial grazing and bacteria to C and N measurements. More important though is showing and addressing that grazer growth rates appear to fluctuate daily (and potentially ingestion rates as well), calling into major question their major conclusions about infected Ehux resulting in stronger grazer growth.


Details on Major issues
1) It was unclear to me from the methods how O. marina growth rates were measured and over what time periods. I assume that these were calculated over the periods indicated in Table 1 (i.e. Tfinal vs T0). A simple line that these were calculated based on t=0 and the final time point (which is what I assume was done) would help clarify. However, the authors also then state in the Methods (lines 170-173) that the periodic addition of prey made it difficult to make direct measurement of prey and predator cell concentrations, which would impact calculation of growth and ingestion rates. I appreciate that the authors clearly state that any cell loss through viral lysis in the Eh-infected conditions makes it hard to actually measure ingestion rates (line 314-320) and therefore may overestimate these rates. But, can we therefore really trust the reported ingestion rates? Which one is it, yes we can reliably measure growth and ingestion rates or can’t we? I would also add that it seems that one could still accurately account for changes in O. marina cell numbers, by accounting for the slight dilution events that occurred when volume was added during periodic prey additions, so it seems like they should be able accurately calculate predator growth rates. Perhaps they did this in the supplemental tables provided, but that wasn’t clear to me.


What I found really helpful was the time vs E. hux cell number plots from experiment 1 in Fig. S2. Why are these not shown, even as supplemental data, for the other experiments as well? Likewise, it would be valuable to see O. marinus numbers plotted over time to get a better overall sense of the results of these experiments, especially in terms of how ingestion rate seems to drop off after the first few hrs in Exp. 1 and how it may vary over days in Exp’s 2-4.

So, this was intriguing me, so I actually went and looked at the data available in the supplemental tables and did some quick and dirty plots. See attached and related comments below.

Related to this, the authors make a statement (lines 279-291) “Compared to experiment 1, the higher total number of ingested E. huxleyi cells, both infected and non-infected, measured in experiments 2 – 4 indicated that O. marina resumed ingestion after 6 h, as prey cells were digested.” The authors have the data (daily time points) to do this. I in fact re-analyzed the data they provided and see that grazer growth rates at least fluctuate daily. One could do this for the ingestion rates, but it wasn’t clear to me from the two table provided which were the appropriate number to look at it.

Similarly, there is the following sentence (lines 281-282): “Also, the relatively low standard deviation values indicated daily ingestion rates were fairly constant from day to day in experiments 2 – 4.” This statement is presumptive and not substantiated. Again, one could calculate these from their data. In theory, the rates in each experiment could vary wildly from day to day, but still the ‘bulk’ ingestion rate (which again I assume is calculated from t=0 and Tfinal [3,5,or days]) could all be very similar between these three experiments.

The grazer cells numbers suggest that ingestion rates probably fluctuate daily (see below). This was nagging me so I plotted daily O. marina abundances and they fluctuate day to day between the treatments, suggesting day variation in grazer growth rate. I did a quick analysis of these growth rates ( ln(Tx/T(x-1)) ) for Exp. 3, and they do in fact fluctuate. So depending on when you ended the experiment, the ‘bulk’ grazer growth rate could be higher or lower for Ehux or infected Ehux. (Note: that there are two tables reporting cells abundance in their supplemental tables. I’m not sure what is the difference between them, so I plotted the data for both.) This fluctuation day to day throws their overall conclusions about differences in grazing rates between non-infected and infected cells into question. I predict that the daily ingestion rates vary daily as well.


2) The time vs cell number plots in Fig. S1C and S2 suggest that the E. hux prey cells used appear to not actually be growing in exponential phase as reported in the methods section. In Fig. S1C, non-infected cells go up briefly then go down and plateau. This suggests that they are actually in stationary phase or were at best harvested in late-log, just before stationary phase. Likewise, in the non-grazing treatments of E. hux in Fig. S2, E. hux slowly declines over time suggesting that the cells are in stationary phase. Or were they diluted and this is some strange effect on the cells from dilution? These data are in conflict with what the authors state about using cells in log phase. From their experience with these conditions (light, temp, media) at what concentrations does E. hux hit stationary phase (i.e. what does a typical growth curve look like)? I think it is important to conduct these experiments in log phase because cell physiology can be altered in stationary phase. At the very least, the authors should explain these odd patterns and revise their statement that the cells are probably in late-log phase or stationary phase.

Again I was curious about this, so I actually plotted the Ehux abundance for the non-grazing controls. For Exp. 1,2, and 4 the Ehux controls are flat or decline, again indicating that the cells are not growing exponentially. This is a major problem here to address.

3) The final major complicating factor that the authors have not addressed is that the cultures are non-axenic. I fully appreciate that it is difficult (and sometimes impossible) to generate axenic cultures of phytoplankton, but it should be discussion at the least. Bacteria are not much smaller than O. marina, and other studies show that O. marina can ingest and grow on bacteria (Quick literature search found this: Roberts et al 2011 "Feeding in the dinoflagellate Oxyrrhis marina: linking behaviour with mechanisms” J. Plankton Research 2011). At the very least, the authors should address in the discussion how growth on bacterial grazing may affect their results and conclusions. In my opinion, they really need to determine bacterial counts to properly assess any potential difference between the virally-infected and non-infected E. hux cultures in terms of bacterial abundance and grazing. Specifically, it is plausible that viral infection releases organic matter that fuels bacterial growth which then increases O. marina growth. This in fact may precisely explain the perplexing results that ingestion rate of E. hux is lower in +virus conditions but O. marina growth rate is higher.

In addition when setting up the stocks of E. hux for the grazing experiments, the Eh+virus contains E. hux lysate (to inoculate with viruses) but this lysate (<0.45 µm) also will contain cell debris that will fuel bacterial growth, and the –virus E. hux is given the same volume of sterile f/2 to make the E. hux concentrations equal. What is the volume of lysate volume added? Even if it is a small volume, the lysate could be chock full of tasty organic material and sufficient to stimulate the bacteria which then subsequently feeds the O. marina. Also, there were daily additions of E. hux cells to maintain the grazer:prey balance, so the addition of E. hux infected cells could inject additional bacteria and/or lysates to fuel bacterial growth.

4) Similarly, the C and N measurements probably reflect those of E. hux cells and bacteria (GF/F filters catch bacteria). These results are suspect as well and probably are not just C and N content of E. hux.

5) If the reason for increased O. marinus growth rate on Ehux+virus cells really is increased nutrition of infected E. hux due to increased N, can the slight increase in N content account for the increased grazer growth rate, even with the decreased ingestion rate based on cell quotas? It seems like one could do a back of the envelope calculation based on cellular N content of O. marinus and E. hux with the measured ingestion and grazing rates. This still doesn’t address where the C is coming from to fuel increase O. marina growth. The C and fatty acid data don’t support higher C content in infected E. hux cells, so the authors don’t have a plausible explanation for how this works mechanistically.

Again, overall, I think the direction and goals of this study are very interesting and important, but the methods and measurements need some major revision, repeating of the experiments, or strong justifications and explanations to account for my concerns above.

Another positive aspects of the study
Lines 420-422. This is a sound and important conclusion, to look again at the Evans and Wilson study.


Other less important issues.
Line 177 “We postulated a starting population size of 6,000 O. marine cells” Didn’t the authors measure the original grazer populations. Perhaps they mean “For these calculations, we set the starting population size at 6,000”

Line 411-413: “Furthermore, several studies have shown that O. marina responds in a similar way to other microzooplankton taxa to various experimental stimuli (Strom et al. 2003a; Strom et al. 2003b; Tillmann 2004).”

This syntax is confusing, I think the authors mean this: “Furthermore, several studies have shown that O. marina responds to various experimental stimuli in similars ways to that of other microzooplankton taxa (Strom et al. 2003a; Strom et al. 2003b; Tillmann 2004).”

Additional comments

Again, I feel their are major issues to address here, but I welcome rebuttal to my issues, especially if my assessment of the data is incorrect. This especially applies to my re-analysis of the cell count data, so I have attached a PDF that shows as best as I could, the data I pulled and used for the plots for verification.

Annotated reviews are not available for download in order to protect the identity of reviewers who chose to remain anonymous.

---

## Round 0.2 · accepted · Accept

Thank you for taking the time to address the reviewers' prior concerns, they are both happy with the revised manuscript. I am accepting the manuscript, but please read the proofs very carefully since the reviewers noted that there are a few sentences missing words and other typos.

# Reviewer 1 ·

Basic reporting

The articles is well written and clearly presented

Experimental design

The experiments were well designed and conducted. All methods used are well described.

Validity of the findings

The data presented derived from well conducted experiments and all results appear to me well interpreted. The conclusions are thus well supported and further discussion provide future avenues of research that will take further the reported significance of the work.

Additional comments

Dear Authors,
thank you for the feedback from the earlier review and the clear improvement of the manuscript by adding a new experiment (exp 5) verifying that Oxyrrhis growth is not enhanced in the presence of bacteria, virus or other molecular by-products of infection. This validates your main findings.
I suggest a final overview of the manuscript as I detected several words that are connected lacking a space between them. Otherwise, all seems to be good.

Reviewer 2 ·

Basic reporting

No comment.

Experimental design

No comment.

Validity of the findings

No comment.